# Oxygen Transfer in Two-Stage Activated Sludge Wastewater Treatment Plants

Maximilian Schwarz *, Justus Behnisch, Jana Trippel, Markus Engelhart and Martin Wagner

Institute IWAR, Technical University of Darmstadt, Franziska-Braun-Str. 7, 64287 Darmstadt, Germany;
j.behnisch@iwar.tu-darmstadt.de (J.B.); j.trippel@iwar.tu-darmstadt.de (J.T.);
m.engelhart@iwar.tu-darmstadt.de (M.E.); m.wagner@iwar.tu-darmstadt.de (M.W.)
* Correspondence: m.schwarz@iwar.tu-darmstadt.de

**Abstract:** Aeration is an energy-intensive process of aerobic biological treatment in wastewater treatment plants (WWTP). Two-stage processes enable energy-efficient operation, but oxygen transfer has not been studied in depth before. In this study, $\alpha$-factors were determined with long-term ex situ steady-state off-gas measurements in pilot-scale test reactors (5.8 m height, 8.3 m³) coupled to full-scale activated sludge basins. A two-stage WWTP with more than 1 Mio population equivalent was studied over 13 months including rain and dry weather conditions. Operating data, surfactant concentrations throughout the two-stage process, and the effect of reverse flexing on pressure loss of diffusers were examined. The values of $\alpha_{mean}$, $\alpha_{min}$, and $\alpha_{max}$ for design load cases of aeration systems were determined as 0.45, 0.33, and 0.54 in the first high-rate carbon removal stage and as 0.80, 0.69, and 0.91 in the second nitrification stage, respectively. The first stage is characterized by a distinct diurnal variation and decrease in $\alpha$-factor during stormwater treatment. Surfactants and the majority of the total organic carbon (TOC) load are effectively removed in the first stage; hence, $\alpha$-factors in the second stage are higher and have a more consistent diurnal pattern. Proposed $\alpha$-factors enable more accurate aeration system design of two-stage WWTPs. Fouling-induced diffuser pressure loss can be restored effectively with reverse flexing in both treatment stages.

**Keywords:** aeration; alpha ($\alpha$); fine-bubble diffusers; high-rate activated sludge systems (HRAS); off-gas; reverse flexing; surfactants; TOC F/M ratio; TOC sludge loading; wastewater treatment





## 1. Introduction

Aeration is an essential process in aerobic biological wastewater treatment. In most wastewater treatment plants (WWTPs), it accounts for more than half of the net energy consumption [1–3]. Engineers rely on technical standards providing design guidelines to properly design aeration systems [4–6]. Various WWTP process configurations are possible depending on wastewater composition and required effluent target. Each process configuration demands individual design considerations for the aeration system. Technical guidelines, therefore, provide $\alpha$-factors to consider inhibiting effects on oxygen transfer in the activated sludge (AS). The $\alpha$-factor determines oxygen transfer efficiency as the ratio of oxygen transfer under process conditions compared to clean water. However, comprehensive research on oxygen transfer in two-stage AS processes is not available. This study provides planners with $\alpha$-factors required for the design of aeration systems in a two-stage configuration. We discuss the impact of stormwater treatment and fluctuations of operating parameters such as TOC F/M ratio on oxygen transfer in the individual treatment stages. Furthermore, surfactant removal within a two-stage process and the effectiveness of reverse flexing to restore pressure loss of diffusers in the different treatment stages are examined.

### 1.1. Energy Efficiency of Two-Stage Activated Sludge Systems

Currently, almost all conventional activated sludge (CAS) wastewater treatment plants operate in an energy-negative mode. When an HRAS system is followed by a second bi-

ological treatment stage (e.g., for nitrogen removal), it can be operated differently than a CAS system [7]. In this case, the first stage can redirect carbon into waste activated sludge (WAS) through biosorption and energy self-sufficiently remove nutrients [8,9]. Liu et al. [8] presented a variety of A-B process designs, and Jimenez et al. [10] described design parameters to optimize carbon redirection. They defined a typical operation range of HRAS systems as SRT < 1 day, HRT ≈ 30 min, DO < 1 mg $O_2 \cdot L^{-1}$, and very high sludge-specific organic loading rates that result in a concentration of influent particulate, colloidal, and soluble chemical oxygen demand (COD) into the WAS through biosorption. This improves direct energy recovery from carbon-loaded sludge through biogas production [11,12]. Moreover, in-plant energy consumption is reduced by lower oxygen demand for aerobic carbon removal and higher overall aeration efficiency [9]. The separation of carbon- and nitrogen-removing biomass potentially reduces overall oxygen supply by more precise aeration control according to the respective biomass's specific oxygen demand [13]. Depending on the wastewater composition, not enough soluble COD to ensure complete denitrification may be a critical limitation of two-stage processes that is aggravated by additional carbon redirection. Therefore, two-stage WWTPs are recommended for high-carbon or low-nitrogen wastewater treatment; alternatively, they require side-stream short-cut nitrogen removal processes (e.g., nitritation–denitritation or partial nitritation–anammox) to decrease carbon requirement of nitrogen removal [8]. Nonetheless, two-stage activated sludge configurations are a sustainable option in the ongoing shift from conventional treatment by removal in WWTPs to more energy-efficient treatment and resource recovery in water resource recovery facilities (WRRFs) [8].

### 1.2. Influences on Oxygen Transfer in Two-Stage Activated Sludge Systems

The $\alpha$-factor is determined as the ratio of volumetric oxygen mass transfer coefficient in process water compared to clean water as described in ASCE 18-18 [14] and DWA-M 209 [15]. The use of a separate fouling factor (F or $\alpha$F) to distinguish between diffuser- and wastewater-specific effects on oxygen transfer is described by EPA [4]. Recent review articles summarize the influences on oxygen transfer in process conditions. Baquero-Rodríguez et al. [3] reviewed a variety of factors including diffuser aging and fouling, influent wastewater variability, and airflow rates for fine-pore diffuser aeration. Amaral et al. [16] focused on the modeling aspect of the gas–liquid transfer in activated sludge. Both studies concluded that the development of a model to consider all factors affecting oxygen transfer in activated sludge systems would be extremely valuable. So far, the complexity of interactions between factors complicates the development of a comprehensive $\alpha$-model. To achieve this goal, more knowledge about the involved processes has to be acquired.

Therefore, one path to gain deeper insight is to look at extreme variations of activated sludge process designs such as two-stage configurations. The biosorption mechanism utilized for carbon redirection in HRAS stages describes surface adsorption of particulate and colloidal organic matter on sludge flocs and storage of soluble COD inside of biomass [17,18]. This can have a positive effect on oxygen transfer as substances inhibiting gas-transfer at the bubble–bulk interface are removed or adsorbed on sludge flocs. Garrido-Baserba et al. [19] discussed strategies to increase oxygen transfer efficiency through biosorption, inter alia by specifically removing surfactants. The amphiphilic structure of surfactants causes a negative effect on oxygen transfer at low concentrations in clean water [20] and activated sludge [21]. High biosorption of surfactants in a first treatment stage could improve oxygen transfer in a subsequent treatment stage. The overall energy efficiency of an aeration system is determined not only by oxygen transfer in the bulk liquid, but also by pressure loss of diffuser elements. This pressure loss resembles the extra resistance that blowers have to overcome to widen membranes and diffuse air through the membrane perforation. Pressure loss increases due to fouling, aging, and scaling of membranes, and it also negatively affects oxygen transfer efficiency [22,23]. Reverse flexing is a mechanical cleaning method where diffuser membranes are relaxed by turning off the blowers and releasing pressure from the air pipes. This causes a rapid collapse of the

diffuser membrane onto the diffuser's frame under hydrostatic pressure. Turning on the blowers flexes the diffuser's membrane and reopens its slits, which removes biofilm and particulate matter from the membrane surface. As a result, previously built-up pressure loss is mitigated which enables more energy-efficient operation of the aeration system [24,25].

*1.3. Goals of This Study*

Factors relevant for energy-efficient operation of aeration systems have been studied for CAS systems; however, comprehensive research is not available for two-stage AS processes. This paper addresses this research gap and defines $\alpha$-factors for design load cases applicable to design aeration systems of two-stage AS systems by measuring oxygen transfer on a pilot scale. Most importantly, the underlying measurements include variations of diurnal cycle of WWTP operation and influent characteristics, rain and dry weather, and seasonal variations affecting oxygen transfer and the $\alpha$-factor. The resultant dataset covers various load cases of a two-stage WWTP. Some procedures to design aeration systems use static $\alpha$-factors, whereas the approach of German guideline DWA-M 229-1 [5], as described in Wagner and Stenstrom [26], distinguishes three load cases with $\alpha_{mean}$, $\alpha_{min}$, and $\alpha_{max}$ factors that we determined accordingly. We also quantified surfactant concentrations in samples throughout the treatment process to examine the distribution of surfactants in the treatment stages of a two-stage configuration. Additionally, different operation of treatment stages within a two-stage system affects bioflocculation capability and resultant sludge composition, which could have an effect on diffuser fouling. We investigated operation and maintenance of fine-bubble diffusers in those conditions through a series of diffuser pressure loss measurements after reverse flexing to determine if fouling can be mitigated effectively in two-stage processes.

## 2. Materials and Methods

*2.1. Design and Operation of Pilot-Scale Test Reactors*

Long-term ex situ steady-state off-gas monitoring was conducted in pilot-scale test reactors as described in ASCE/EWRI 18-18 (2018). Tank dimensions were 1.2 m $\times$ 1.2 m $\times$ 5.8 m (L $\times$ W $\times$ H) with a volume of 8.3 m$^3$. Two reactors were operated to examine both AS stages of a two-stage process in parallel. Both reactors were equipped with fine-bubble disc diffusers (ELASTOX-T EPDM TYP B, WILO GVA, Wülfrath, Germany) with a diffuser density of 13.5%. Unlike off-gas measurements using off-gas hoods, the airflow rate within an ex situ reactor can be varied independently from the operation of the WWTP it receives its sludge from. A range of airflow rates (specified for aerated tank volume—$q_{Vol,aer}$) between 0.75 and 2.25 Nm$^3 \cdot$m$^{-3} \cdot$h$^{-1}$ was set, covering typical ranges of two-stage WWTPs. Sludge transfer pumps (AGNM02 NEMO$^{®}$, NETZSCH Holding, Selb, Germany) were operated to maintain a constant hydraulic retention time (HRT) of 15 min as recommended by ASCE/EWRI 18-18. Sludge flow was measured with electromagnetic flowmeters (Promag W 400, Endress + Hauser AG, Reinach, Switzerland). Mixing conditions within the tanks can be assumed close to an ideal continuous stirred tank reactor (CSTR), because of the combined energy input of aeration and sludge transfer.

Mean values of clean water tests of standard oxygen transfer rate (SOTR) were used as a denominator for the $\alpha$-factor. Clean water tests were conducted with electrochemical dissolved oxygen (DO) probes (Oxymax COS51D, Endress + Hauser AG, Reinach, Switzerland) with a fast response time $t_{90}$ of 30 s. Slower optical DO probes Oxymax COS61D, Endress + Hauser AG, Reinach, Switzerland) were used in process conditions, as long-term testing did not require a fast response time, and their lower maintenance offers more reliable DO measurement in activated sludge operation. While off-gas measurements require a steady inflow, clean water tests were conducted without continuous inflow. In our pilot plant lateral sludge inflow improved oxygen transfer at low airflow rates, which resulted in overestimates of the $\alpha$-factor. As a consequence, only airflow rates above 0.75 Nm$^3 \cdot$m$^{-3} \cdot$h$^{-1}$ were considered in this study. Clean water tests were conducted before and after a long-term off-gas measurement period to evaluate diffuser conditions.

This revealed a decrease in SOTR of 2–6% depending on airflow rate and a dynamic wet pressure increase of about 1 kPa. These results primarily indicate inevitable aging of diffusers and secondarily indicate scaling and fouling. Overall, the effect of scaling and fouling during the long-term off-gas measurement was kept low due to monthly pressure cleaning and reverse flexing of disc diffusers twice a week. Therefore, in this study, the oxygen transfer is reported as an α-factor instead of an αF-factor. Additionally, potential biofilm build-up on the reactor tank walls was prevented with monthly cleaning and visual inspection to ensure only suspended biomass transferred from the adjacent full-scale AS tanks was examined in the ex situ reactors for off-gas measurements. Online sensors were cleaned twice a week to prohibit solids deposition and biofilm growth affecting optical instruments.32op

Other parameters and their sensors and instruments for off-gas measurements were airflow rate measured with thermal mass flowmeters (Proline t-mass A 150, Endress + Hauser AG, Reinach, Switzerland), off-gas concentrations of oxygen (paramagnetic sensor) and carbon dioxide (NDIR) measured with a gas-analyzer (X-STREAM Enhanced, Emerson Electric Co., MO, USA) that receives dry off-gas free of particles (CSS-V, M&C TechGroup, Ratingen, Germany), atmospheric pressure (Cerabar PMC21, Endress + Hauser AG, Reinach, Switzerland), atmospheric temperature (Omnigrad T TST434, Endress + Hauser AG, Reinach , Switzerland), and electrical conductivity (Indumax CLS50D, Endress + Hauser AG, Reinach, Switzerland). Data were recorded in 30 s intervals by online sensors and summarized as 15 min averages. This resulted in high-resolution data that matched the HRT of the test reactors and the interval of operating data provided by the WWTP operator. However, residence time distribution in an ideal CSTR yields a 63% replacement of activated sludge in the reactors at HRT of 15 min and 98% at 1 h, respectively. Therefore, for final analysis, 1 h intervals were composed to prevent autocorrelating observations. In total, α-factors were recorded for 9 months in long-term off-gas measurements covering a period of 13 months.

### 2.2. Design and Operation of Examined Two-Stage WWTP

The examined two-stage activated sludge WWTP has a design capacity of more than 1 Mio PE. It has a mean dry weather influent flow of 2.6 $m^3 \cdot s^{-1}$ and a maximum wet weather influent flow of almost 7 $m^3 \cdot s^{-1}$ of mostly municipal wastewater, complying with German effluent standards. Raw wastewater is first treated in screens (width 10 mm) and an aerated grit chamber before it flows into the primary clarifier with a mean HRT of 60 min that ranges from 35 to 100 min depending on influent flow. Biological wastewater treatment is split into a first high-rate activated sludge stage for carbon removal and a subsequent second stage for nitrification with a fivefold larger tank volume. Both aerated stages are plug flow reactors with tapered aeration, while 25% of tank volume of the second stage is a continuously mixed upstream denitrification zone. Both treatment stages have no internal recirculation and are followed by clarifiers that return activated sludge into the respective stages. A bypass line can pass 0.2 $m^3 \cdot s^{-1}$ of primary effluent into the second stage to redirect organic carbon required for biological nutrient removal in the upstream denitrification. A recirculation line can recirculate 0.5 to 0.55 $m^3 \cdot s^{-1}$ of nitrate containing final clarifier effluent into the first stage. This relieves the final downstream denitrification (biofiltration) which removes remaining nitrate. These concepts are described in more detail in Jimenez et al. [10] and Wandl et al. [27].

Influent wastewater load is diluted in activated sludge tanks and the concentration of removable substances changes within AS tank zones during treatment, especially in plug flow reactors. Therefore, determining the α-factor of a whole plug flow reactor tank at a certain time requires off-gas testing across all subsequent aeration zones [28,29]. However, to closely monitor the diurnal cycle of oxygen transfer, activated sludge was transferred from the front aeration zone of both plug flow aerated stages into the pilot-scale test reactors. Operating data of the first and second stage of the examined two-stage activated sludge WWTP are summarized in Table 1.

**Table 1.** Operating data of examined two-stage WWTP.

| Parameter (Abbreviation) | Unit | First Stage | | Second Stage | |
|---|---|---|---|---|---|
| | | Mean ± SD | 5th–95th Percentile | Mean ± SD | 5th–95th Percentile |
| Volume specific airflow rate ($q_{Vol,aer}$) | $Nm^3 \cdot m^{-3} \cdot h^{-1}$ | 1.8 ± 0.5 | 0.9–2.3 | 0.7 ± 0.2 | 0.5–1.0 |
| Dissolved oxygen (DO) | $mg \cdot L^{-1}$ | 0.6 ± 0.3 | 0.2–1.0 | 3.2 ± 0.2 | 3.0–3.4 |
| Actual hydraulic retention time ($HRT_a$) | h | 0.7 ± 0.1 | 0.5–0.9 | 1.9 ± 0.3 | 1.4–2.4 |
| Nominal hydraulic retention time ($HRT_n$) | h | 2.0 ± 1.0 | 0.9–3.6 | 6.2 ± 2.6 | 2.8–10.5 |
| Sludge retention time (SRT) | d | 1.9 ± 0.7 | 0.7–3.2 | 31 [1] ± 7.3 | 21–44 |
| Total solids in AS (TS) | $g \cdot kg^{-1}$ | 3.0 ± 0.4 | 2.4–3.7 | 6.1 ± 0.6 | 5.1–7.1 |
| Volatile fraction in AS (MLVSS/TS) | % | 72 ± 6 | 63–85 | 59 ± 4 | 53–65 |
| TOC inflow concentration ($TOC_{in}$) | $mg \cdot L^{-1}$ | 75 ± 22 | 44–113 | 18 ± 5.4 | 12–25 |
| Water temperature ($T_W$) | °C | 17 ± 3 | 13–22 | 17 ± 3 | 13–22 |
| Total suspended solids in effluent ($TSS_{effluent}$) | $mg \cdot L^{-1}$ | 25 ± 12 | 12–46 | 4.1 ± 1.7 | 2.1–7.6 |
| Sludge volume index (SVI) | $mL \cdot g^{-1}$ | 99 ± 35 | 51–164 | 49 ± 5.5 | 41–56 |

Note 1: Median value; for all other parameters the median deviates by less than 10% from the above-listed means.

The sample standard deviation marks the dispersion from mean values during standard operation of the WWTP, while the 5th and 95th percentiles are stated to describe reasonable minimum and maximum operation conditions that are only exceeded in exceptional cases. Volume specific airflow rate $q_{Vol,aer}$ is specified in relation to aerated basin volume. The reported sludge retention time is temperature-corrected to 15 °C (correction coefficient = 1.072, compare Clara et al. [30]), and outliers outside 1.5 times the interquartile range above and below Q1 and Q3 quartiles were removed. A rolling mean was calculated of the remaining data spanning 2 days for the first stage and 30 days for the second stage. These chosen timespans resemble the median SRT in the respective stages. Online turbidity sensors (SOLITAX sc, Hach Lange GmbH, Düsseldorf, Germany) measuring mixed liquor suspended solids are calibrated for total solids (TS) and regularly compared with laboratory analysis (according to EN 12880). Mixed liquor suspended solids (MLSS) are not measured regularly. On average, MLSS was 0.8 $g \cdot L^{-1}$ lower than TS. Total organic carbon (TOC) inflow concentration ($TOC_{in}$) considers all inflows of a treatment stage (e.g., supernatant of return activated sludge and bypass flows) proportional to their respective water flow. This combination is required because effluent TOC of the intermediate clarifier recycled into the first stage with return activated sludge has a share of about 30% of total TOC inflow in the first stage. TOC concentrations are measured by ex situ online analyzers (QuickTOC, LAR, Berlin, Germany) in the influent and effluent of the first stage and drift-corrected to match laboratory analysis (EN 1484). We used TOC as a suitable sum parameter to describe influent wastewater characteristics instead of COD, because ex situ online analyzers of TOC are common in larger WWTPs and enable an analysis with higher temporal resolution than COD laboratory analysis. For comparison, TOC/COD ratios based on laboratory analysis were 0.33 ± 0.05 in the influent of the first stage and 0.46 ± 0.10 in the influent of the second stage. $TSS_{effluent}$ is recorded in the supernatant of the respective clarifier (2 μm pore size). Hydraulic retention time (HRT) refers to the retention time in activated sludge tanks, not the whole treatment stage with clarifiers. It is stated either as nominal $HRT_n$ which considers only influent flow or as actual $HRT_a$, which includes recirculation flows, as well as main wastewater inflow (compare nomenclature in Henze et al. [31]). The TOC F/M (feed to mass) ratio is typically derived from TOC concentration in the inflow, MLSS in the AS, and volume of biological treatment stage. To account for dilution in the AS tank and return TOC load of recirculation flows, we use the volume proportional $TOC_{in}$ and $HRT_a$ as described above. To simplify comparison, TS is assumed as given in units of $g \cdot L^{-1}$ similar to MLSS. Thus, we derived an actual TOC $F/M_a$ ratio from parameters given in Table 1 as follows:

$$\text{TOC F/M}_a \text{ ratio} = TOC_{in} \cdot TS^{-1} \cdot HRT_a^{-1} \ (kg \cdot kg^{-1} \cdot day^{-1}). \tag{1}$$

The use of actual $HRT_a$, volume proportional $TOC_{in}$, and resultant TOC $F/M_a$ ratio reflects organic load in the AS tanks more reasonably regarding their effect on oxygen transfer in the front aerated zones than the nominal $HRT_n$ and TOC F/M ratio.

### 2.3. Separate Rain and Dry Weather Conditions

This study distinguished rain and dry weather conditions to examine their impact on oxygen transfer in the AS tanks. WWTP operators typically record all-day weather conditions; however, these do not reflect the diurnal inflow dynamic. Instead, we assigned a weather category on the basis of the diurnal variations of collected inflow data. Figure 1 shows the inflow course during diurnal cycle as smoothed functions of percentiles of the inflow represented by the lines. Top and bottom lines describe the percentiles at 0% and 100%, while the lines in between depict percentiles from 5% to 95% in 10% steps. The dashed line serves as a distinction where data above were assigned as rain and data below were assigned as dry weather category. It represents the 80th percentile of inflow data based on recorded weather conditions. In the operating data of the examined two-stage WWTP, 77% of days were recorded as dry weather (dry and frost conditions), while the remaining 23% were recorded as rain weather (e.g., rainfall, snowfall, and discharge from stormwater retention basins). Therefore, the 80th percentile was chosen to clearly separate rainfall periods from regular operation. A wastewater inflow of $3 \, m^3 \cdot s^{-1}$ is considered as rain weather at 6:00 and as dry weather at 12:00. The 85th and 95th percentiles are categorized as rain weather but have a distinct diurnal inflow pattern. While a single rainfall runoff does not follow this pattern, on average, light rainfall is added on the dry weather pattern. In contrast, the 100th percentile represents maximum inflow capacity of the WWTP and is constant throughout the diurnal cycle.

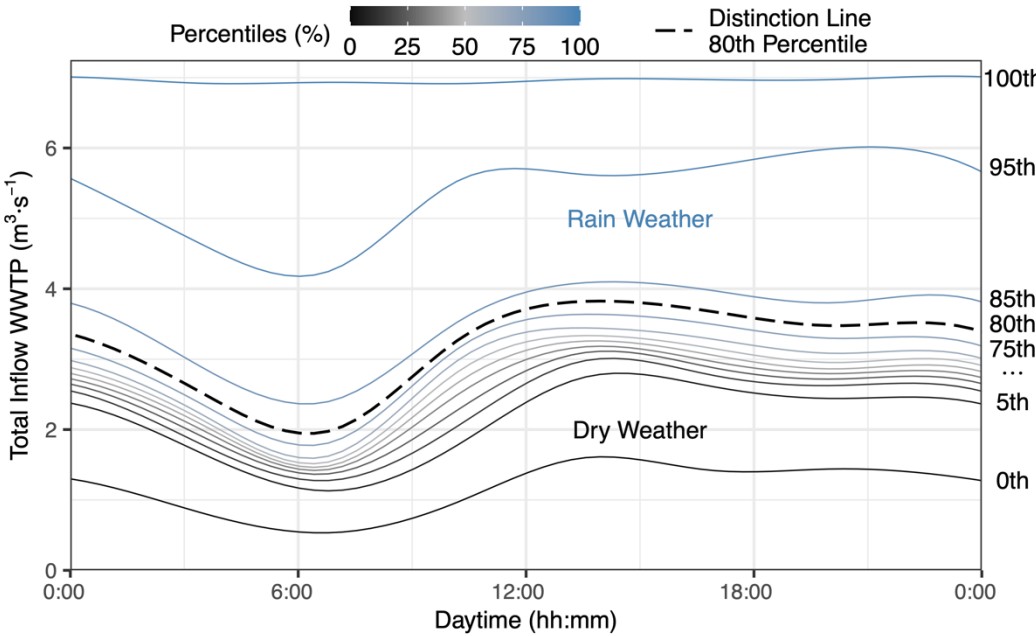

**Figure 1.** Assigning weather category on the basis of diurnal variations of total wastewater inflow.

### 2.4. Surfactant Analysis

Surfactant concentrations of successive treatment stages of the two-stage AS WWTP were measured with Hach cuvette tests (Hach Lange GmbH, Düsseldorf, Germany) for anionic (LCK 332), nonionic (LCK 333), and cationic surfactants (LCK 331) in a spectrophotometer (DR 3900, Hach Lange GmbH, Düsseldorf, Germany). Accordingly, 24 h composite samples were taken from primary clarifier influent, first-stage influent, second-stage influent, and second-stage effluent. Grab samples were taken from the first- and second-stage

activated sludge tank and settled before taking an aliquot from the supernatant to analyze. The samples were not centrifuged or filtered. However, when taking an aliquot, intake of particles was avoided. Overall, surfactant cuvette tests are error-prone because other surfactant types may cause low-bias results according to working procedure information by the manufacturer. Duplicate measurements of each sample with a recovery were conducted according to the manufacturer's working procedure. The measurement series was repeated three times over the course of 1 year. In total, at least five evaluable tests per surfactant type are available for each sample location with a recovery between 80% and 120%.

### 2.5. Dynamic Wet Pressure Measurement and Reverse Flexing Procedure

Dynamic wet pressure (DWP), also known as pressure drop, pressure loss, or diffuser headloss, is the pressure difference of a submerged diffuser calculated as the difference between pressure in the air pipe close to the diffuser and the hydrostatic pressure. DWP increases with higher airflow rates; therefore, it is usually specified at a specific airflow rate. Pressure was measured with a capacitive digital pressure transmitter in the air pipes close to the diffuser frame (Cerabar PMC21, Endress + Hauser AG, Reinach, Switzerland). DWP was calculated as the difference of this sensor reading and the hydrostatic pressure in the reactor defined by blow-in water depth, which is limited by an overflow in the test reactors.

Reverse flexing was performed twice a week during maintenance of the pilot reactors, which resulted in a period of 3 to 4 days since the last procedure. To perform reverse flexing, blowers were shut off for up to 2 h and relative pressure in the air pipes was reduced to 0 kPa. The diffusers remained sealed during the long-term measurements as no water leakages were detected in the diffuser frame. Because DWP increases with airflow rate, long-term measurement series were conducted at a constant airflow rate for better comparison. Activated sludge from the first stage was aerated at constant airflow rate of 1.5 and 1.9 $Nm^3 \cdot m^{-3} \cdot h^{-1}$, and sludge from the second stage was aerated at constant airflow rate of 0.8 and 1.0 $Nm^3 \cdot m^{-3} \cdot h^{-1}$. Tests at lower airflow rates were run for 36 days and those at higher airflow rates were run for 26 days. Diffusers were cleaned with high pressure before each measurement series.

## 3. Results and Discussion

### 3.1. Effect of Rainfall and Diurnal Cycle on Oxygen Transfer

The oxygen transfer in the AS process is subject to a multitude of influence factors that vary seasonally and within daily cycles. Additionally, hydraulic and organic loading differ tremendously between rain and dry conditions, thus affecting oxygen transfer in the activated sludge tanks. Table 2 presents all α-factors measured within this study as described in Section 2 for mean ± standard deviation and 5th and 95th percentiles.

**Table 2.** α-factors determined with ex situ off-gas measurements in a two-stage WWTP.

| Parameter (Abbreviation) | Unit | First Stage | | Second Stage | |
|---|---|---|---|---|---|
| | | Mean ± SD | 5th–95th Percentile | Mean ± SD | 5th–95th Percentile |
| α-Factor (ex situ measurement) | - | 0.43 ± 0.06 | 0.33–0.54 | 0.80 ± 0.07 | 0.69–0.91 |

Figure 2A divides α-factors by treatment stage and weather conditions in an empirical cumulative distribution. The horizontal dashed lines mark the 5th and 95th percentiles. Lower mean α-factors were measured in the first stage (0.43) than in the second stage (0.80), as indicated by the vertical dashed lines. Kroiss and Klager [9] stated similar α-factors of 0.45 and 0.7 in first and second stages of the Vienna main wastewater treatment plant. Overall, influences affecting oxygen transfer differ tremendously between the first and second stage in a two-stage AS configuration. In particular, the first high-rate stage

cannot be compared with CAS systems, where $\alpha$-factors for systems with nitrification and denitrification typically fall into the range of 0.6 to 0.75 [32]. Additionally, the distinction of rain and dry weather reveals that $\alpha$-factors in the first stage decreased during high inflows of rainwater, whereas no such effect was apparent in the second stage. The effect of stormwater runoff on oxygen transfer has not been discussed in the literature so far. However, rain events have an impact on multiple parameters potentially affecting oxygen transfer in the activated sludge tank, as shown before. Stormwater runoff affects the hydraulic and influent load of a WWTP. A first flush often brings a high load due to washout of sewer sediments followed by slightly contaminated rainwater afterward [33]. Wilén et al. [34] concluded that biological processes in the sewer system are more aerobic at high flows and more anaerobic at low flows, thus changing wastewater properties. Typical effects of rain events also include lower conductivity and water temperature with increased total inflow (data not shown), which is compensated for by standardization to norm conditions when determining $\alpha$-factors.

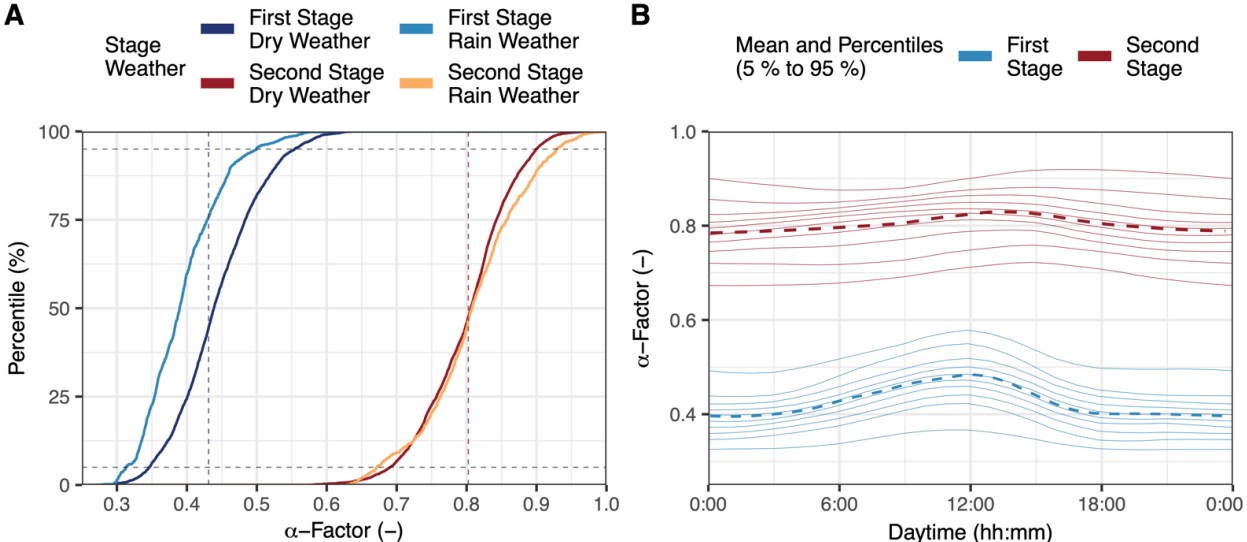

**Figure 2.** Empirical cumulative distribution (**A**) and diurnal variation (**B**) of $\alpha$-factors as percentiles (solid lines) and means (dashed lines) in the examined two-stage AS WWTP.

Figure 2B shows the diurnal variation of recorded $\alpha$-factors in both stages. The lines represent the course of percentiles from 5% to 95% as described in Section 2 for Figure 1. The first stage was characterized by a distinct peak of the $\alpha$-factor at noon, regularly fluctuating between 0.39 and 0.48, as indicated by the dashed line representing the mean $\alpha$-factor. Peak $\alpha$-factors are measured during daytime instead of nighttime due to a long retention time of wastewater in a large sewer system. In contrast, $\alpha$-factors in the second stage had a smoother course without a distinct peak. Here, $\alpha$-factors fluctuated on average between 0.78 and 0.83 within a day. The influent load into the second stage was decreased and buffered by the preceding HRAS tank and upstream denitrification zone, resulting in a smoother diurnal cycle of $\alpha$-factors. This also explains the different extent of rain effects on oxygen transfer in two-stage AS treatment stages, as further discussed below.

The diurnal cycle of $\alpha$-factor observed in the first stage was previously described by an inverse relationship of $\alpha$-factor and influent load [35,36]. For operators of WWTPs, this negative correlation means that oxygen transfer is generally at its lowest when oxygen demand is highest. To illustrate this relationship, Figure 3 displays the volume specific airflow rate ($q_{Vol,aer}$) in the full-scale AS tanks as the dependent variable of TOC inflow concentrations ($TOC_{in}$) and $\alpha$-factor. Blowers were controlled by DO in the aeration basins to set the airflow rate. First, Figure 3A shows that volume specific airflow rate was increased in response to higher $TOC_{in}$ to meet resulting oxygen demand of biomass in

both stages. Secondly, lower $\alpha$-factors forced operators to increase airflow rates to meet this oxygen demand, as shown in Figure 3B. This figure also reveals that this relationship was more distinct in the first stage than the second. The two stages also differed during rain weather, where lower $\alpha$-factors coincided with higher airflow rates in the first stage, but no significant decrease in $\alpha$-factor was apparent in the second stage. It is important to note that $\alpha$-factor is usually not affected by airflow rate directly, but rather coincides with changes in oxygen demand due to influent load [29,37].

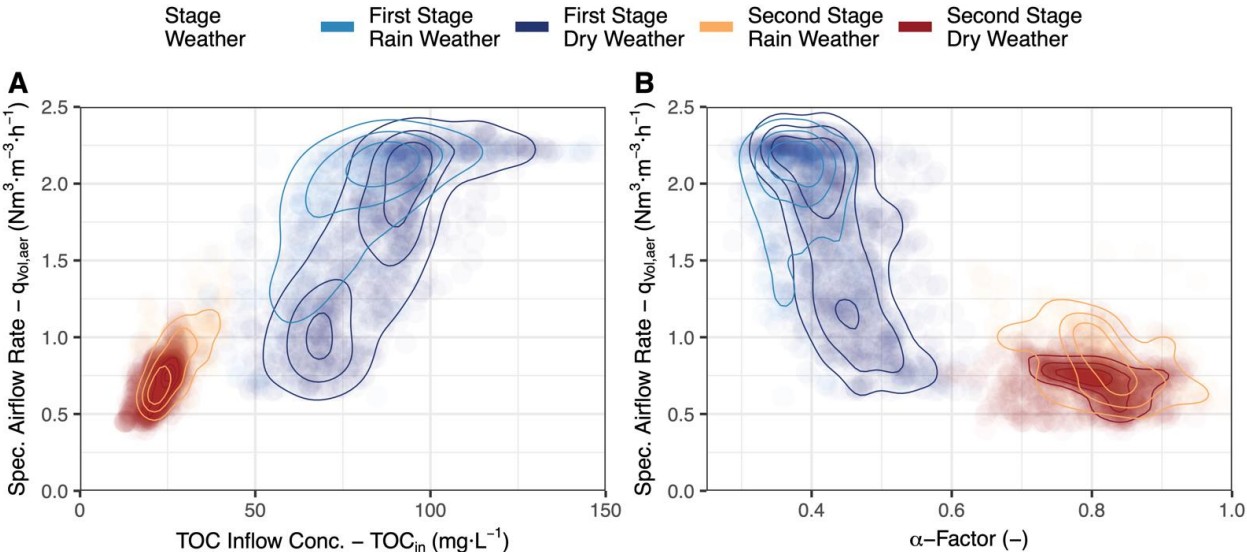

**Figure 3.** Volume specific airflow rate of full-scale aeration basins for $TOC_{in}$ (**A**) and $\alpha$-factor (**B**) grouped for treatment stages and weather conditions.

In Figure 3, the individual points represent mean data recorded within 1 h intervals. Colors distinguish between rain and dry weather periods as specified in Section 2. To visualize the two-dimensional distribution of the resulting clusters, they were divided by three density lines with each interval containing 25% of the respective cluster data. A smaller area enclosed within these density lines denoted a higher density of the contained data points.

Overall, these results show that oxygen transfer in the second stage was more stable than in the first stage. It is important to emphasize the resultant effect on the required airflow rate to meet oxygen demand in the treatment stages; the described daily fluctuation of $\alpha$-factor from 0.48 to 0.39 in the first stage required an increase of 22% of the airflow rate to compensate for oxygen transfer inhibition. In comparison, a decrease from 0.83 to 0.78 in the second stage required adjustment of airflow rates of only 6% within a typical day. Moreover, Table 2 and Figure 2 reveal the range and distribution of potential $\alpha$-factors in the two stages caused by various influences on oxygen transfer.

### 3.2. Influence of Organic Loading on Oxygen Transfer

Below, we further examine influences that resulted in the presented range of $\alpha$-factors. The TOC F/M ratio is a suitable aggregate parameter that correlates with oxygen transfer inhibition [38]. Figure 4 displays four scatterplots of measured $\alpha$-factors for TOC F/$M_a$ ratio and its individual components: actual hydraulic retention time ($HRT_a$), TOC inflow concentration ($TOC_{in}$), and total solids (TS).

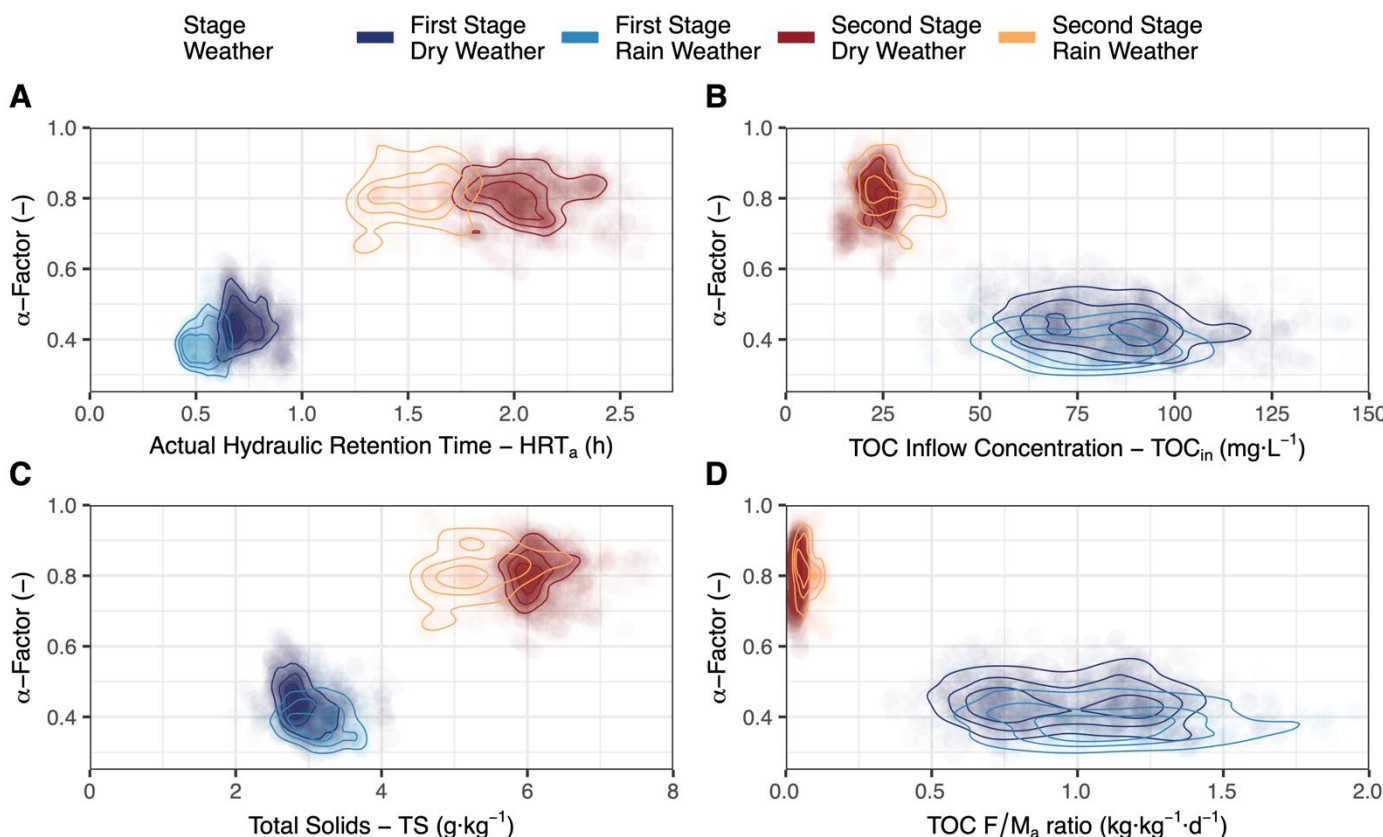

**Figure 4.** The $\alpha$-factors for $HRT_a$ (**A**), $TOC_{in}$ (**B**) TS (**C**), and the aggregated parameter TOC F/$M_a$ ratio (**D**) grouped for treatment stages of a two-stage WWTP and weather conditions.

Figure 4A shows the $\alpha$-factors recorded in the first and second treatment stages at their respective $HRT_a$. The treatment stages of the examined two-stage WWTP were operated differently and, as a result, all diagrams in Figure 4 clearly distinguish both stages from each other. Moreover, rain and dry weather categories were clearly separated within treatment stages, as $HRT_a$ reflects high and low water inflow. Overall, lower $\alpha$-factors were recorded in the first treatment stage with its shorter $HRT_a$. The longer $HRT_a$ within the first stage indicated slightly higher $\alpha$-factors, while no such effect could be seen in the second stage. Although water inflow and the resultant $HRT_a$ have no known direct impact on oxygen transfer, a change of hydraulics in a WWTP affects other parameters that have an impact on the $\alpha$-factor.

TOC inflow concentrations in the first stage were higher and spread over a wider range than in the second stage, as displayed in Figure 4B. Roughly two-thirds of TOC influent load was removed in the first stage. While Figure 3 suggests a clear correlation between $\alpha$-factor and TOC influent concentration, Figure 4B shows that it was less evident within the respective treatment stages. However, looking at both treatment stages, a negative correlation between TOC inflow concentration and oxygen transfer can still be confirmed. Jiang et al. [36] concluded a similar negative logarithmic relationship between $\alpha$ and COD on the basis of measurements in three WWTPs. Ahmed et al. [39] applied a power function to fit an $\alpha$-model for SBR reactors. Both approaches came to similar results to this study but examined different WWTP process configurations that are not directly comparable to the examined two-stage process. The major difference between $\alpha$-factors in treatment stages can be attributed to the oxygen transfer inhibiting characteristics of readily biodegradable substrate [39], especially accumulation of surfactants on the bubble surface [32,40]. During rain periods, $\alpha$-factors observed in the first treatment stage were lower than in dry conditions, although TOC inflow concentrations were similar or lower. However, TOC load increased when considering the increased water flow and organic load

of a first flush in the sewer system as a result of a rainfall event, thus explaining lower $\alpha$-factors. This effect was not apparent in the second stage. Here, TOC inflow concentration was slightly higher during rainy weather as some organic load remained untreated at low $HRT_a$ in the first stage. Nonetheless, $\alpha$-factors in the second stage did not decrease because most influent organic load was buffered in the first stage and the upstream denitrification zone of the second stage. The implementation of an upstream denitrification stage has been reported as advantageous for oxygen transfer in CAS systems [41]. Thus, the high $\alpha$-factors in the second stage can be attributed in part to this, even though some readily biodegradable substrate was passed into the second stage by the bypass line.

Figure 4C displays $\alpha$-factors for total solids (TS). The $\alpha$-factors and TS in the second stage were higher than in the first stage and high for activated sludge process in general. Within the second stage, no correlation with TS was indicated, while a slight decrease in $\alpha$ was apparent in the first stage, coinciding with rain weather. This outcome is discussed in more detail in Section 3.3.

The TOC $F/M_a$ ratio in Figure 4D combines the previously discussed parameters. Its course was similar to $TOC_{in}$ in Figure 4B, except for the first stage during rainfall events. Here, high water inflow and TOC concentration produced higher TOC $F/M_a$ ratios with a negative effect on $\alpha$-factor. Günkel-Lange [38] examined the relationship between COD F/M ratio and $\alpha$-factor for extended aeration, nitrogen-removal, and carbon-removal CAS systems and proposed an inverse linear correlation. Again, the examined two-stage WWTP is different from CAS systems and complicates direct comparison. However, the presented data complement the understanding of oxygen transfer dynamics in more complex WWTP process configurations.

According to the diagrams in Figure 4, oxygen transfer in the second treatment stage was seemingly unaffected by any variation of the presented parameters. However, this cannot be concluded from the above analysis with certainty, as at most only two interactions were taken into account in each diagram. Furthermore, the combined parameter TOC $F/M_a$ ratio obscured variation of its individual components (e.g., 100 kg/h TOC load at 3 g/L TS would result in the same TOC F/M ratio as 200 kg/h TOC load at 6 g/L TS, but the resulting conditions would affect oxygen transfer differently). Considering both treatment stages, our results confirm the inverse relationship between $TOC_{in}$ or TOC $F/M_a$ ratio and $\alpha$-factor, as presented in previous studies. However, no single parameter illustrated in Figure 4 correlated significantly with the $\alpha$-factor when considering oxygen transfer in individual treatment stages.

*3.3. Interaction of Suspended Solids and Hydraulic Load with Oxygen Transfer*

Generally, TSS concentration, usually measured as mixed liquor suspended solids (MLSS), inversely correlates with the $\alpha$-factor. This has been extensively demonstrated for membrane bioreactors (MBR), where different rheology of thick sludge at MLSS up to 30 g·L$^{-1}$ has an influence on gas transfer dynamics [42–44]. Henkel [45] proposed that the volatile fraction of suspended solids (mixed liquor volatile suspended solids—MLVSS) in particular causes oxygen transfer inhibition. These studies extrapolated the inverse relationship measured in MBRs into conventional activated sludge systems (CAS), where typical MLSS concentrations are below 6 g·L$^{-1}$. In contrast, newer studies stated that biosorption decreases the concentration of organic substances in the soluble phase, thereby reducing oxygen transfer and inhibiting accumulation in the gaseous phase [39,46]. Higher MLSS increases the biosorption of organic matter in CAS, which additionally improves carbon redirection in HRAS stages [10,47]. As a consequence of biosorption as the dominant impact on oxygen transfer, a positive correlation between MLSS concentrations up to 6 g·L$^{-1}$ and $\alpha$-factor was proposed by Baquero-Rodríguez et al. [3]. Overall, there seems to be no robust relationship between MLSS and $\alpha$-factor for CAS [39]. Modeling $\alpha$ from MLSS does not include possible influences of floc structure on oxygen transfer, which vary inevitably between WWTPs. It is probable that floc size (e.g., measured as particle size distribution), settling characteristics (SVI), or addition of precipitants (e.g., for phosphorus

removal) alter the liquid–solid interface, thus also influencing the gas–liquid and gas–solid interfaces. To summarize, MLSS or TS as typical parameters in wastewater treatment cannot describe all properties of the solid and liquid phase that are relevant to the dynamic of oxygen transfer once the gas phase is added.

Below, we discuss various parameters to describe the solid and liquid phase in the treatment stages of the examined two-stage WWTP and their potential influence on the $\alpha$-factor. As shown in Figure 4C, total solids were overall higher in the second stage ($6.1 \pm 0.6$ g·kg$^{-1}$) than in the first stage ($3.0 \pm 0.4$ g·kg$^{-1}$). In contrast, the volatile fraction of the respective sludges was higher in the first stage ($72 \pm 6\%$) than in the second stage ($59 \pm 4\%$). Although Henkel [45] argued that the inverse relationship between the $\alpha$-factor and the solid phase is better described by MLVSS than MLSS, this is not immediately obvious when comparing the absolute MLVSS in the two-stage WWTP. Here, MLVSS was still higher in the second stage (~3.6 g·L$^{-1}$) than in the first stage (~2.2 g·L$^{-1}$), even though $\alpha$-factors were higher in the second stage. Thus, in our results, a potential negative effect of organic content of sludge measured as MLVSS was superimposed by enhanced biosorption in the second stage, ultimately increasing oxygen transfer. This is supported by various characteristics that could be beneficial to oxygen transfer in the second stage compared to its preceding first stage, such as better sludge settling (SVI of $49 \pm 5.5$ mL·g$^{-1}$ compared to $99 \pm 35$ mL·g$^{-1}$). This would also result in lower total suspended solids in effluent ($4.1 \pm 1.7$ mg·L$^{-1}$ in second stage instead of $25 \pm 12$ mg·L$^{-1}$ in first stage). The activated sludge was also altered by addition of sodium aluminate as precipitant for phosphorus removal in the influent and effluent of the second stage. Overall, this also affected the liquid phase, which had a visually distinguishable higher turbidity of supernatant from the first-stage activated sludge compared to the clear supernatant of sludge samples from the second stage. SVI, TS$_{effluent}$, precipitant use, or turbidity of supernatant have not previously been used to explain oxygen transfer in the AS process. Their individual influence on oxygen transfer cannot be quantified, because only two stages with opposed characteristics were examined in our study. However, these parameters further describe characteristics of the solid phase within the two-stage process that could explain the overall difference of $\alpha$-factors between the first and second stage.

Within the second stage, no correlation of $\alpha$-factor with TS was indicated, whereas a slight decrease in $\alpha$ was apparent in the first stage, coinciding with rain weather, as depicted in Figure 4C. Rainfall affected TS concentrations differently in the treatment stages of the two-stage WWTP. Figure 5A illustrates the relationship between TS and HRT$_a$ for both rain and dry weather inflow in the respective treatment stage. At lower HRT$_a$ and high hydraulic load during rainy weather, TS decreased in the second stage, while it remained stable in the first stage. This is unexpected as processes with higher HRT and SRT are generally less susceptible to biomass washout due to stormwater flows [48,49]. Examining operating data indicated that this may have been caused by washout of TS from the primary clarifier into the first stage at shorter HRT$_a$ (data not shown). However, the elevated TS concentrations might not have been the only cause of lower $\alpha$-factors during stormwater treatment in the first stage. HRT$_a$ represents the possible adsorption contact time of soluble and colloidal organic substances with sludge flocs within the AS tank. Once this organic load is adsorbed on sludge flocs, it is removed through waste activated sludge in the clarifier, and it is also less likely to inhibit oxygen transfer in the gas phase. Jimenez et al. [10] determined optimal operating conditions of an HRAS system (260 L, CSTR) for removal of soluble, colloidal, and particulate COD at HRTs of >15 min, >30 min, and >45 min, respectively. As a conclusion, low HRT$_a$ caused by rainwater inflow decreased biosorption capacity in the first stage which left more soluble and colloidal organic substances that could accumulate in the gas phase, thus decreasing the $\alpha$-factor. On the contrary, the $\alpha$-factor did not drop at lower HRT$_a$ and TS in the second stage (see Figure 3A,C). However, as the second stage received low organic load (see Figure 3B), biosorption mechanisms most probably were much less pronounced than in the first stage.

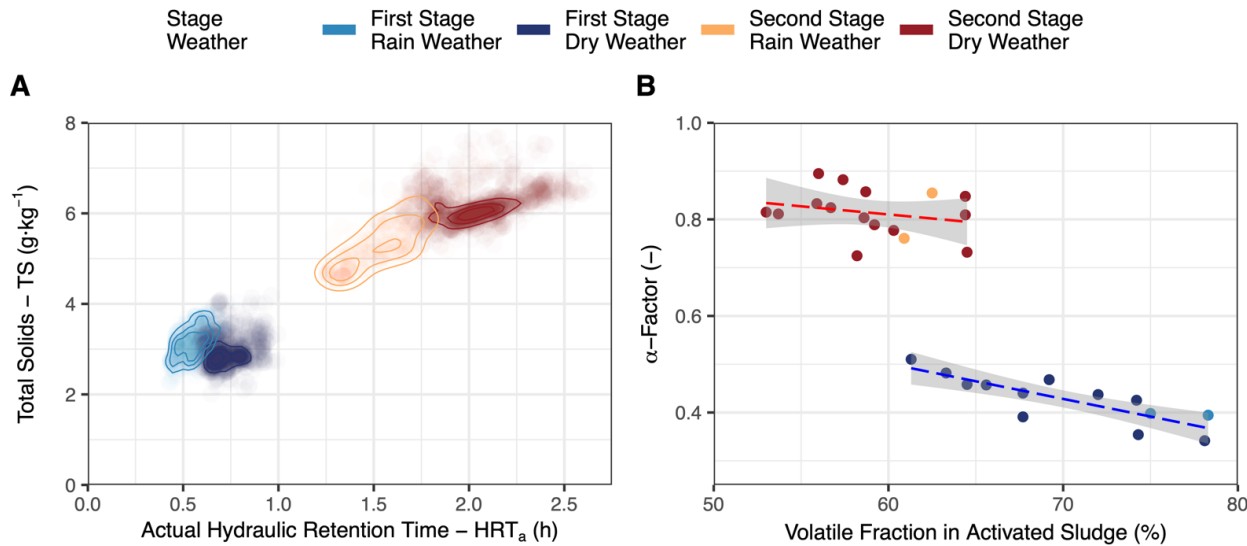

**Figure 5.** TS for HRT$_a$ (**A**) and $\alpha$-factor for volatile fraction in activated sludge as daily mean (**B**), grouped for both treatment stages of a two-stage WWTP and weather conditions.

The $\alpha$-factors are summarized as daily mean values in Figure 5B and compared with volatile fraction determined from grab samples of activated sludge. Overall, the volatile fraction was lower in the second stage than the first stage due to lower load, higher SRT, and the addition of sodium aluminate as precipitant for phosphorus removal. Within the treatment stages, the regression line surrounded by 95% confidence intervals revealed a negative correlation of $\alpha$-factor with volatile fraction in the first stage. While an effect potentially remained, no significant correlation was apparent in the second stage. Operating data revealed a slightly elevated volatile fraction in activated sludge, as well as return activated sludge, at lower HRT$_a$ (data not shown), which could have further decreased $\alpha$-factor during rainy weather. As a conclusion, the suggested negative correlation of $\alpha$-factor with the volatile fraction of solids by Henkel [45] is one of the mechanisms determining oxygen transfer dynamics within the first stage. A stronger impact of volatile fraction was demonstrated in the first stage, whereas, in the second stage, it was superimposed by other influences.

It is worth mentioning that the individual impact of wastewater parameters on $\alpha$-factor discussed in this study cannot be derived and quantified from the above analysis. In contrast to a controlled experimental design in which all examined parameters are varied systematically, we measured oxygen transfer of an operating full-scale WWTP. The resulting dataset describes only a combination of parameters occurring in real conditions. Additionally, building a mechanistic model of influences on oxygen transfer with a multivariate analysis produces unreliable results when based only on two AS stages that are operated as differently as in the examined two-stage WWTP. The diagrams in Figure 3 show no overlap between $\alpha$-factors measured in the treatment stages and their process parameters. Hence, complementing our results with further data from CAS systems is necessary to fill these gaps and enable more general inference from wastewater treatment parameters on oxygen transfer dynamics. Lastly, although treatment capacity and overall oxygen demand certainly change throughout seasons, no strong seasonality of $\alpha$-factor can be derived from our results thus far. Nonetheless, our results allow a complete assessment of $\alpha$-factors for aeration system design purposes in a two-stage WWTP.

*3.4. Design Load Cases for Aeration Systems of Two-Stage WWTPs*

The design of aeration systems of WWTPs specifies the number of diffusers and airflow rates to meet oxygen demand in activated sludge tanks. Diffuser manufacturers state standardized oxygen transfer parameters determined in clean water. However,

to consider oxygen transfer inhibition occurring in activated sludge, these parameters have to be multiplied by the $\alpha$-factor. This design process has been described in various technical guidelines and reference books [4,6,50]. Oxygen transfer inhibition depends on the WWTP's treatment goal and various processes, among other factors. However, no $\alpha$-factors have been proposed for two-stage WWTP process configurations thus far. Therefore, according to our results from long-term measurements, we propose $\alpha$-factors for the design of aeration systems in two-stage systems.

The design approach of German standard DWA-M 229-1 [5], based on Günkel-Lange [38], applies mean, minimum, and maximum $\alpha$-factors to define load cases. The $\alpha_{mean}$ represents the average operation conditions of a WWTP. We, therefore, calculated $\alpha_{mean}$ as the average of all $\alpha$-factors measured during dry weather operation at the examined two-stage WWTP that fell between the mean $\pm$ standard deviation of $HRT_a$, TS, and $TOC_{in}$, as stated in Table 1. From this, we derived $\alpha_{mean}$ values of 0.45 and 0.80 for the first and second stages, respectively. Because no rainy weather was considered for $\alpha_{mean}$, it was slightly higher than the average of all measurements in the first stage (0.43), while there was no difference in the second stage (0.80, compare Table 2). The $\alpha_{min}$ and $\alpha_{max}$ values describe oxygen transfer inhibition during high and low load of the WWTP, respectively. We defined these $\alpha$-factors on the basis of a comprehensive dataset including seasonal variation, as well as rain and dry weather conditions, measured within a 13 month period of conducting long-term off-gas measurements. Hence, we approximated $\alpha_{min}$ and $\alpha_{max}$ as the 5th and 95th percentiles of the full dataset, respectively. These percentiles were chosen with a remaining measurement uncertainty in mind. If the design process requires otherwise, the full set of measured data is shown in Figure 2. Our proposed $\alpha$-factors to design aeration systems in two-stage configurations are summarized in Table 3. These results are applicable for the design of aeration systems in two-stage WWTPs similar to the one examined in this study.

**Table 3.** The $\alpha$-factors for design load cases of two-stage activated sludge WWTPs.

| Treatment Stage | $\alpha_{mean}$ (−) | $\alpha_{min}$ (−) | $\alpha_{max}$ (−) |
| --- | --- | --- | --- |
| First stage (HRAS) | 0.45 | 0.33 | 0.54 |
| Second stage | 0.80 | 0.69 | 0.91 |

*3.5. Removal of Surfactants in Two-Stage WWTPs*

Surfactants have a negative effect on oxygen transfer even at low concentrations due to their amphiphilic structure. They adsorb on the gas–liquid interface of bubbles, as well as on the solid phase of sludge flocs and other particles. Quantifying surfactant loads throughout the wastewater treatment process allows identifying which treatment stage is particularly affected by oxygen transfer inhibition and which treatment process eliminates surfactants. Although a decrease in surfactant concentrations with each treatment stage is expected, the extent of such a reduction is not obvious in two-stage configurations. Effluent quality of a HRAS stage is poor because it is followed by a second treatment stage. First-stage settling tank effluent is characterized by a visible turbidity, remaining mean TOC of 48 mg·L$^{-1}$, and $TSS_{effluent}$ of 25 mg·L$^{-1}$ (see Table 1). Thus, the remaining surfactant concentration passing into the second stage cannot be neglected for oxygen transfer and has to be measured.

Figure 6 shows boxplots of surfactant concentrations of successive treatment stages of the examined two-stage WWTP divided into three surfactant types. The median of each surfactant type in a sample is summed and connected by a dashed line (median total). Boxplots and the trendline show that surfactant concentrations decreased throughout the treatment stages. Most importantly, total surfactant concentration decreased about 70% from first-stage influent to second-stage influent, and a dilution of influent concentration in both treatment stages was apparent, as concentrations in the activated sludge supernatant were lower than the preceding influent concentrations. Anionic and nonionic surfactants

were more prevalent in the samples, which is typical for municipal wastewater composition [51,52]. Although absolute concentrations of individual cuvette tests are unreliable, the performed measurement series provides a reasonable span of concentrations for each treatment stage. In comparison, Odize [46] measured anionic surfactants in HRAS influent ($8 \pm 2$ mg·L$^{-1}$) and effluent ($1 \pm 0.1$ mg·L$^{-1}$), both of which are within the above described surfactant concentration range. The overall surfactant removal of more than 95% within the WWTP is in line with other studies [51]. The high surfactant concentrations measured in the first treatment stage correspond to low $\alpha$-factors ($0.43 \pm 0.06$), as well as lower surfactant concentrations and higher $\alpha$-factors ($0.80 \pm 0.07$) in the second stage. Hence, the previously described higher alpha values in the second stage can also partially be attributed to the adsorption and biological removal of surfactants in the first stage.

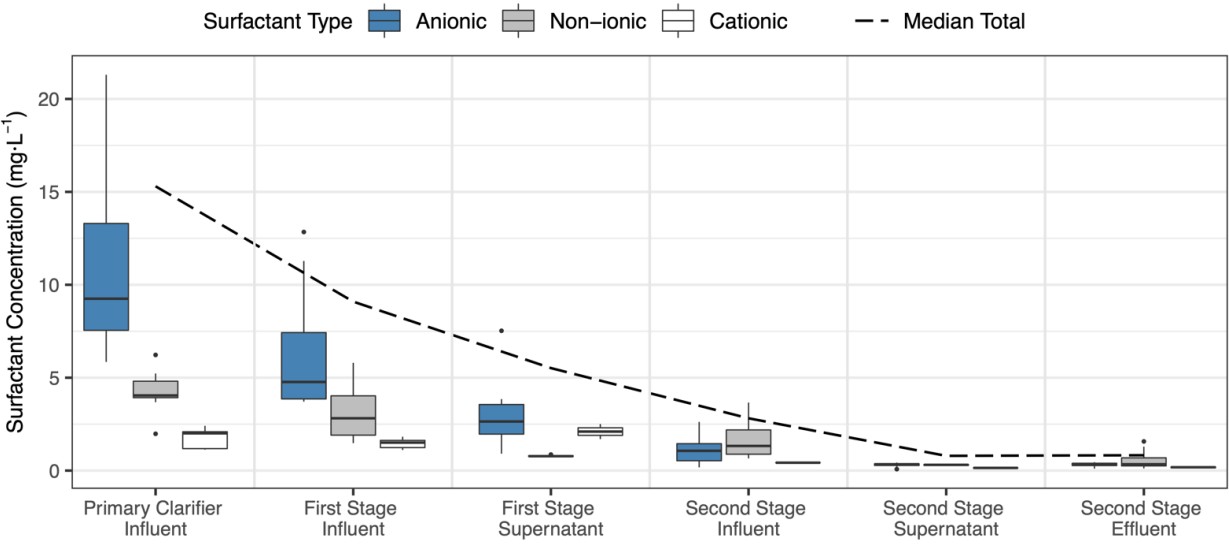

**Figure 6.** Surfactant concentrations of successive treatment stages divided into surfactant types.

### 3.6. Reverse Flexing in Two-Stage Processes

Influencing factors on fouling in biological wastewater treatment have been studied extensively for membrane bioreactors [53], whereas the effect of fouling on diffuser membranes has focused primarily on quantifying economic implications [22,54,55]. Knowledge about site-specific wastewater characteristics and WWTP operation on fouling of diffuser membranes is sparse. Thus, Rosso et al. [56] even suggested implementing on-site long-term column testing of various diffusers as part of the design procedure to take site-specific fouling effects into account when selecting diffusers. As discussed before, inflow wastewater characteristics in the treatment stages of the examined two-stage WWTP and their operation differ; therefore, sludge characteristics differ as well. The resulting separated biomasses with higher content of heterotrophic organisms in the first stage for high-rate carbon removal and autotrophic organisms in the second stage for nitrification could affect fouling behavior of diffusers differently. So far, it is unknown whether existing diffuser maintenance procedures can be applied to mitigate the pressure loss of diffusers in two-stage WWTPs.

Figure 7 shows the boxplots of measured DWP within 12 h intervals after reverse flexing was performed. Median values revealed an expected increase of DWP within the typical 3.5 day interval between maintenance. Most interquartile ranges spanned less than 1 kPa of DWP difference except the test series in the first stage at 1.9 Nm$^3$·m$^{-3}$·h$^{-1}$, where airflow rate fluctuated by $\pm 0.5$ Nm$^3$·m$^{-3}$·h$^{-1}$ due to blower limits. Within the test series, no systematic increase in DWP during multiple cleaning intervals was observed (data not shown), which would be expected over longer periods without periodic pressure cleaning [23,24]. According to these test series, we can conclude that pressure loss can be

restored effectively with reverse flexing in both treatment stages of a two-stage WWTP. In conclusion, operators of a two-stage WWTP do not have to adapt different diffuser maintenance intervals or procedures for the two treatment stages.

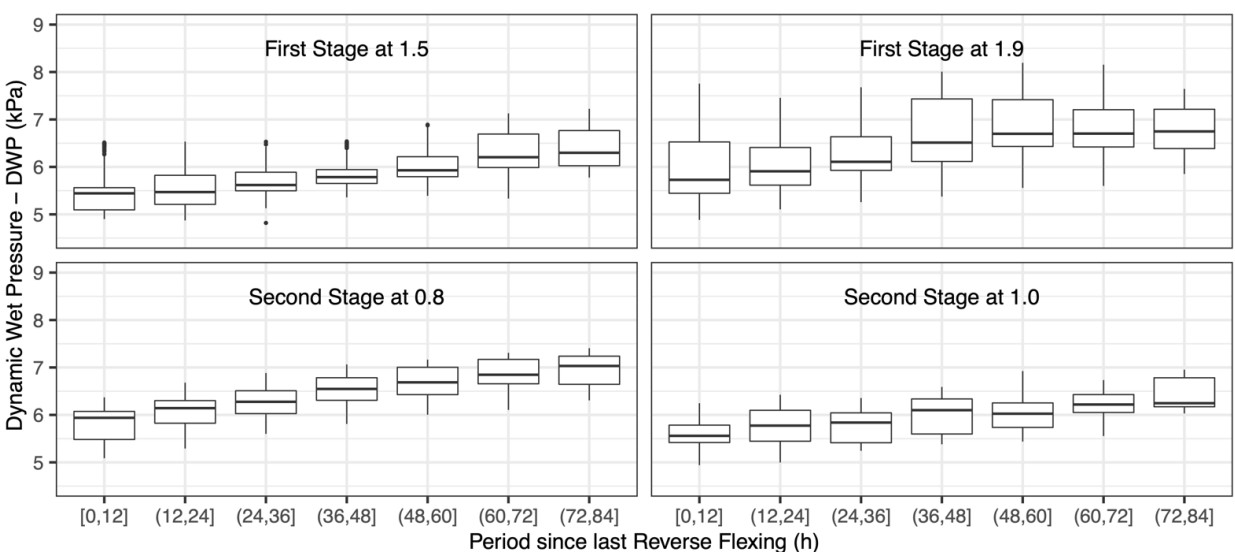

**Figure 7.** Increase in DWP of disc diffusers since last reverse flexing procedure during operation in activated sludge from first and second stage and at two specific airflow rates ($Nm^3 \cdot m^{-3} \cdot h^{-1}$).

## 4. Conclusions

On the basis of our long-term off-gas measurements, we summarize below our findings relevant for design and operation of aeration systems in two-stage activated sludge WWTPs.

1.  This paper defined $\alpha$-factors for the first and second stages of a two-stage WWTP. The underlying off-gas measurements on a pilot scale covered a typical range of operation conditions of such a process, as detailed in Table 1, including seasonal variation, as well as dry and wet weather conditions. As a result, $\alpha$-factors for design load cases were derived for practical application to design aeration systems more accurately. They were determined as 0.45 for $\alpha_{mean}$ and 0.33/0.54 for $\alpha_{min}/\alpha_{max}$ in the first stage (HRAS), and as 0.80 for $\alpha_{mean}$ and 0.69/0.91 for $\alpha_{min}/\alpha_{max}$ in the second stage. Because different process configurations of two-stage processes exist, these $\alpha$-factors can be transferred to configurations similar to the one examined in this study. No range of $\alpha$-factors for two-stage processes was previously proposed.

2.  Our results show how key operating parameters influence the oxygen transfer in the activated sludge system. Most importantly, the impact of high TOC concentrations in inflow resulting in lower oxygen transfer rates can be confirmed and quantified for a two-stage activated sludge process. TS and $HRT_a$ in the treatment stages were affected differently by stormwater treatment. As a result, $\alpha$-factor decreased in the first stage, whereas the second stage remained unaffected during high wastewater inflow. Hence, engineers can more accurately decide whether an aeration system design meets the demands of a similar WWTP to that examined in this study. Nonetheless, individual wastewater parameters cannot describe $\alpha$-factor due to various interacting influences. Therefore, applying machine learning methods to predict oxygen transfer is a multivariate approach that we will examine in the future.

3.  Inflow surfactant concentrations measured in 24 h composite samples revealed that surfactant load was significantly lower in the second stage compared to the first stage. Surfactants had a disproportionate influence on oxygen transfer compared with TOC.

Lower α-factors in the first stage could be attributed to this effect but not quantified specifically for surfactants compared to TOC in general.

4. The positive effect of reverse flexing as a maintenance method to restore dynamic wet pressure was observed in both stages. There was no significant difference in fouling effect on diffusers, although sludge composition differed tremendously between the high rate and nitrification stage. Therefore, operators of two-stage WWTPs do not have to adapt different maintenance intervals when planning a reverse flexing schedule.

**Author Contributions:** Conceptualization, M.S.; methodology, M.S. and M.W.; validation, M.S., J.B., J.T., M.E., and M.W.; formal analysis, M.S.; data curation, M.S.; writing—original draft preparation, M.S.; writing—review and editing, M.S., J.B., J.T., M.E., and M.W.; visualization, M.S.; supervision, M.E. and M.W.; project administration, M.S. and M.W.; funding acquisition, J.B., M.S., and M.W. All authors read and agreed to the published version of the manuscript.

**Funding:** We thank the German Federal Ministry of Education and Research (BMBF) for funding the research project WOBeS "Advanced optimization of aerations systems: Investigation for increase efficiency of fine bubble diffusers through adapted process engineering and operational management" (Research Grant 02WA1461).

**Institutional Review Board Statement:** Not applicable.

**Informed Consent Statement:** Not applicable.

**Data Availability Statement:** Measurement data presented in this study are available on request from the corresponding author. Operating data from WWTP operators are not publicly available.

**Acknowledgments:** We acknowledge support from the Deutsche Forschungsgemeinschaft (DFG—German Research Foundation) and the Open Access Publishing Fund of Technical University of Darmstadt.

**Conflicts of Interest:** The authors declare no conflict of interest. The funders had no role in the design of the study; in the collection, analyses, or interpretation of data; in the writing of the manuscript, or in the decision to publish the results.

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
