# Peer review of "Oxygen Transfer in Two-Stage Activated Sludge Wastewater Treatment Plants"

_water, doi:10.3390/w13141964_

Round 1

Reviewer 1 Report

This paper by Schwarz et al describes comprehensively the analysis of a 2-stage biological process. This configuration, initially developed and promoted in Germany, has been recently enjoying a revival in interest worldwide. I read the article with pleasure. It is well written and the work is thorough. The paper presents valuable plots and ranges of alpha that can be used by design engineers, for both the cases of dry and wet weather.

Some comments in detail are reported below.

Abstract

The statement about oxygen transfer not being studied previously, for 2-stage processes, should be amended as “not studied in depth”. After all, the authors acknowledge and cite the previous work, albeit limited in scope, performed by colleagues at DC Water.

Methods

Line 181. Is the flow split or fed in series to the two stages?

Table 1. The SRT of the second stage is very long and this should be the reason for such high alpha values there. Also, I imagine that the authors mean to report the MLSS when they list the TS in AS. In this row, the second stage exhibits 5-7 grams per litre of biomass. In general, this would be considered difficult or impossible to separate in the secondary settler, especially in winter. The authors should elaborate on the effects of these high solids against oxygen transfer (see my comment below).

Discussion

Line 313. I concur that the effects of stormwater runoff on alpha are undocumented.

Fig. 2. I particularly like the plots in Fig. 2 because they provide a much-needed design range to design engineers. I suggest a dimensionless time for Fig. 2b (e.g., t / tpeak), since different plants experience peaks at different time of day, but the pattern remains the same overall.

Fig. 3. I suggest adding a brief description of the density lines in the caption.

Line 451. I agree that floc morphology and size distribution are key element to be included in a thorough analysis of the effect of solids on alpha factors. I think Stenstrom and collaborators have studied size distribution and its effects.

Line 607. I could not agree more with this statement. It is indeed sparse the knowledge about site-specific conditions.

The authors cite the work by Henkel, who studied the effects of biomass on oxygen transfer. My understanding is that biomass does apply a negative effect on oxygen transfer, at higher concentrations. This effect is additive to the process loading. Consequently, would the alpha values for the second stage need to be adjusted downwards? I am most concerned about the last value, exceeding 0.9, which is uniquely high. Despite the likely absence of organics this far long in the process, I imagine that the high MLSS would still act as detriment to oxygen transfer. I encourage the authors to elaborate more on this point.

References

There are a couple of references of theses or dissertations that may not be as easy to retrieve as indexed papers. I suggest choosing indexed papers, whenever the information is also available there.

There is a 2nd version of the IWA Biological Treatment Book, perhaps that would be the up-to-date citation to use.

Author Response

Thank you very much for your review! I really appreciated that someone with knowledge in the field of aeration has spent the time to review my work. Please find my answers to your questions below. 

Best regards from Darmstadt, Maximilian Schwarz 

Abstract

Q: The statement about oxygen transfer not being studied previously, for 2-stage processes, should be amended as “not studied in depth”. After all, the authors acknowledge and cite the previous work, albeit limited in scope, performed by colleagues at DC Water.

A: Added “in-depth” in the sentence.
You are correct: both the DC Water publications on carbon redirection and the modifications on the Vienna main WWTP are cited in our study. Because none focus on oxygen transfer, your suggested wording is more precise.  

Methods

Q: Line 181. Is the flow split or fed in series to the two stages?

A: The biological stages are mostly fed in series. A small share of primary treatment effluent bypasses the first stage and is directed into the second stage as described in line 185-186.

Q: Table 1. The SRT of the second stage is very long and this should be the reason for such high alpha values there.

A: Agreed. There is a very useful paper by Rosso et al., 2005 on this topic (Fifteen Years of Offgas Transfer Efficiency Measurements on Fine-Pore Aerators: Key Role of Sludge Age and Normalized Air Flux). In the paper, a large dataset of off-gas measurements reveals the positive relationship between α-factor and SRT, but also shows its large variance. It is a useful parameter to look at when comparing many different WWTP process layouts. The reason we did not discuss SRT in more detail in this publication is that it is a nasty parameter to calculate properly and combines several sludge and process characteristics in one number:

  • Properly determining the SRT when WWTP operators discontinuously withdraw waste activated sludge is almost impossible, see also Balbierz 2017 (Comparison of methods for solids retention time determination and control). Especially when the goal is to compare SRT with other data recorded in a 1-h interval. In this case, some sort of data pre-processing to smoothen SRT values would be required that would bloat our paper at best or skew the data at worst.
  • SRT is a great thumb parameter for operators and easy to display in a paper. However, as SRT combines many sludge and WWTP process characteristics in one number, some valuable information is lost. We therefore think a future approach to discuss (or predict) oxygen transfer relates it to several parameters (some of which are also included in SRT) that can be tracked by already available sensors.

Q: Also, I imagine that the authors mean to report the MLSS when they list the TS in AS.

A: We reported TS (total solids) instead of MLSS or TSS (total suspended solids) because sensors by the WWTP operator were calibrated on TS. Unfortunately, there are no regular laboratory MLSS comparison values available. Our own measurements showed that MLSS is on average 0.8 g/L lower than TS in both stages. I added this information in the methods section. In the table I added the information that Volatile Fraction in AS is calculated as the MLVSS/TS-ratio.

Q: In this row, the second stage exhibits 5-7 grams per litre of biomass. In general, this would be considered difficult or impossible to separate in the secondary settler, especially in winter. The authors should elaborate on the effects of these high solids against oxygen transfer (see my comment below).

A: 5-7 g TS/L equates to about 4-6 g TSS/L. Nonetheless, settling of the high solids concentration is improved by addition of a precipitant for phosphorus removal.  

Discussion

Q: Line 313. I concur that the effects of stormwater runoff on alpha are undocumented.

A: We also had discussions on how to describe potential effects of stormwater runoff on oxygen transfer. Unfortunately, our paper can only point out that some wastewater properties change (see cited literature line 319) that could have an impact on oxygen transfer.

Q: Fig. 2. I particularly like the plots in Fig. 2 because they provide a much-needed design range to design engineers.

A: Thank you!

Q: I suggest a dimensionless time for Fig. 2b (e.g., t / tpeak), since different plants experience peaks at different time of day, but the pattern remains the same overall.

A: While a dimensionless abscissa would be preferable when comparing different WWTPs, in this case the relative positioning between the two treatment stages would be lost. Although it is not a distinct diurnal peak, the second stage appears to peak a little later than the first stage. To not lose this information, I did not change the diagram.

Q: Fig. 3. I suggest adding a brief description of the density lines in the caption.

A: I would also like to do so to improve interpretability, but the MDPI water editorial guidelines suggest using single-line captions. A detailed description is given in line 358-363.

Q: Line 451. I agree that floc morphology and size distribution are key element to be included in a thorough analysis of the effect of solids on alpha factors. I think Stenstrom and collaborators have studied size distribution and its effects.

A: You probably refer to the paper by Li and Stenstrom, 2017 (Impacts of SRT on Particle Size Distribution and Reactor Performance in Activated Sludge Processes) which concludes that SRT and particle size distribution have a positive relationship. Unfortunately, no study examined impact of particle size distribution on oxygen transfer so far, so we did not cite one in our paper. Some further remarks:

  • Properly determining particle size distribution of activated sludge flocs is itself a challenge, because available methods break up larger flocs in the process. Also sampling, transport and storage of samples is an issue.
  • Describing floc morphology with a microscope is a tedious process that results in a categorical classification, which is also hard to standardize.
  • I expect advances on this topic could be made once metadata of available turbidity sensors is used or these types of optical sensors become more advanced. Data from different scattering angles of optical sensors could indicate in more detail when a certain particle size range is more prevalent in activated sludge. Although I don’t expect this information to be as accurate as a particle size distribution from laboratory analysis, it would be easier to collect and standardize a methodology.

Q: Line 607. I could not agree more with this statement. It is indeed sparse the knowledge about site-specific conditions.

A: … and although Garrido-Baserba and others have made major contributions on the topic, the link between fouling and diffuser material also requires more research.

Q: The authors cite the work by Henkel, who studied the effects of biomass on oxygen transfer. My understanding is that biomass does apply a negative effect on oxygen transfer, at higher concentrations. This effect is additive to the process loading. Consequently, would the alpha values for the second stage need to be adjusted downwards? I am most concerned about the last value, exceeding 0.9, which is uniquely high. Despite the likely absence of organics this far long in the process, I imagine that the high MLSS would still act as detriment to oxygen transfer. I encourage the authors to elaborate more on this point.

A: Thus far, it is not clear whether the effects of solids/biomass and process loading on oxygen transfer are always additive (and whether this follows a linear relationship). We discuss their interaction (biosorption) based on the cited literature in section 3.3. Henkel proposed a linear relationship between ML(V)SS and α-factor based on studies where he only varied MLSS and not wastewater load. Additionally, he used a lab-scale membrane bioreactor (1 m3) with a plate module of pore size 0.1 µm for sludge separation. It is unclear if sludge floc morphology in these experiments is directly comparable to regular activated sludge systems with clarifiers and therefore whether Henkels model is generally applicable for all activated sludge systems. Baquero-Rodríguez et al. (2018, A Critical Review of the Factors Affecting Modeling Oxygen Transfer by Fine-Pore Diffusers in Activated Sludge) even come to the opposite conclusion that up to a MLSS of about 5 g/L a positive relationship with the α-factor might exist. Overall, we don’t have a complete model of the impact of solids and soluble substances on oxygen transfer as already discussed at length in section 3.3. We therefore propose that our measured values should be seen as an addition to the ongoing discussion without elaborating even further in this section of the paper.

References

Q: There are a couple of references of theses or dissertations that may not be as easy to retrieve as indexed papers. I suggest choosing indexed papers, whenever the information is also available there.

A: Unfortunately, the technical guidelines on aeration technology by the DWA (German water association) are not available in English. The German dissertation by Tobias Günkel-Lange is referenced as a contribution to the German technical guidelines and is available online for free.

Q: There is a 2nd version of the IWA Biological Treatment Book, perhaps that would be the up-to-date citation to use.

A: True, unfortunately it is not available to me yet. The first edition is cited to refer to nomenclature used in this fundamental book, so currency is less important.

Reviewer 2 Report

In my opinion, the paper is a significant contribution to the topic, but some questions should be considered before its acceptation.

A process diagram of the plant studied must be included to make clear the order of the process (line 179), the recirculation flows and the addition of reagents  (line 475).

It is not clear the recirculation from the CAS clarifier into the first stage. Does this mean that  there is a mixture of both biomass? In line 612, the authors indicates separated biomass.

In table 1, it would be interesting to include the ratio COD / TOC. Taking into account that COD is widely used in the real world of wastewater treatment plants, it may be difficult for the reader  to  interpret the results only with the TOC values.

In table 2, it would be interesting  to know the value of α-factor in the second stage working only with primary clarifier without first stage, in order to know the improvement that the first stage entails.

It is unclear to  me what has been the WWTP where the study was conducted.  

Author Response

Thank you very much for your review! Please find our answers to your questions below. 

Q: A process diagram of the plant studied must be included to make clear the order of the process (line 179), the recirculation flows and the addition of reagents  (line 475).
It is not clear the recirculation from the CAS clarifier into the first stage. Does this mean that  there is a mixture of both biomass? In line 612, the authors indicates separated biomass.

A: Thank you for these hints! We added further explanations in the text to clarify. In our opinion an additional process diagram would unnecessarily bloat the paper. More important than the specific process layout of the examined WWTP are the resultant conditions in the activated sludge as described by the parameters that we discuss oxygen transfer with in chapter 3. Because every WWTP is unique, the calculation of these parameters is described in detail and readers can follow these descriptions for varying layouts to compare with our results.

Q: In table 1, it would be interesting to include the ratio COD / TOC. Taking into account that COD is widely used in the real world of wastewater treatment plants, it may be difficult for the reader  to  interpret the results only with the TOC values.

A: Great advice! We added a description of the TOC/COD ratios in the influent of the treatment stages in the methods section.

Q: In table 2, it would be interesting  to know the value of α-factor in the second stage working only with primary clarifier without first stage, in order to know the improvement that the first stage entails.

A: This was not possible in a fully operational large-scale WWTP. We are currently examining a CAS system for further comparison though.

Q: It is unclear to  me what has been the WWTP where the study was conducted.  

A: We agreed with the WWTP operator to anonymize any findings of our studies. In our opinion the exact location of the WWTP is not relevant for the publication.

Reviewer 3 Report

This problem is relevant for journal scope. 
The concept and aim are clearly defined. I could find some typing errors. The topic of the article is up to date, the introduction and literature survey is easy detailed, the authors discussed already all available literature sources. The presentation and discussion of the result is clear and very detailed. The conclusions are well extracted from the results and discussion. The manuscript follows the formal regulations of MDPI journals.
I suggest the acceptance after minor revision.

Remarks and suggestions
Please cite more papers from MDPI journals at the last 2-3 years in the similar topic of this research.
Write your keywords in alphabetical order
Check the grammar of the English language

Author Response

Q: Please cite more papers from MDPI journals at the last 2-3 years in the similar topic of this research.

A: So far, relevant studies in aeration technology were published mostly in IWA and ASCE journals. We do not select citations based on journal but on relevance for the discussion. With this paper we want to bring relevant research in the field of aeration technology to an MDPI journal.

Q: Write your keywords in alphabetical order

A: Done, thank you for the editorial advice.

Q: Check the grammar of the English language

A: Also done, we had another round of minor corrections.
